# P2 Receptors: Novel Disease Markers and Metabolic Checkpoints in Immune Cells

**DOI:** 10.3390/biom12070983

**Published:** 2022-07-14

**Authors:** Valentina Vultaggio-Poma, Francesco Di Virgilio

**Affiliations:** Department of Medical Sciences, University of Ferrara, 44121 Ferrara, Italy; vltvnt@unife.it

**Keywords:** T cells, T-cell metabolism, P2 receptors, extracellular ATP, inflammatory disease, tumour microenvironment

## Abstract

Extracellular ATP (eATP) and P2 receptors are novel emerging regulators of T-lymphocyte responses. Cellular ATP is released via multiple pathways and accumulates at sites of tissue damage and inflammation. P2 receptor expression and function are affected by numerous single nucleotide polymorphisms (SNPs) associated with diverse disease conditions. Stimulation by released nucleotides (purinergic signalling) modulates several T-lymphocyte functions, among which energy metabolism. Energy metabolism, whether oxidative or glycolytic, in turn deeply affects T-cell activation, differentiation and effector responses. Specific P2R subtypes, among which the P2X7 receptor (P2X7R), are either up- or down-regulated during T-cell activation and differentiation; thus, they can be considered indexes of activation/quiescence, reporters of T-cell metabolic status and, in principle, markers of immune-mediated disease conditions.

## 1. Introduction

T lymphocytes (T cells) are key components of the adaptive immune response and are endowed with the ability to migrate across tissues, ready for pathogen or tumour-antigen recognition. Antigen binding drives T-cell activation and differentiation into specific effector cells with distinct functions and ability to secrete cytokines and other molecular mediators, thus affecting neighbouring and distant cells. Two major subsets of T cells are known: CD8^+^ T cells, mainly specialized in cytotoxic effector functions; and CD4^+^ T cells, also designated as T helper (Th) cells, which differentiate into many effector subsets [1]. The most frequent CD4^+^-T-cell subtypes are: interferon-gamma (IFN-ɣ)-secreting Th1 cells, typically involved in chronic inflammation and anti-viral immunity; Th2 cells, which support allergic responses and immunity against metazoan parasites (e.g., *Ascaris lumbricoides*) and also promote immunosuppression; Th17 cells releasing IL-17 in response to fungal or bacterial infection and promoting autoimmunity [2]; Treg cells that suppress effector-T-cell functions, dampening inflammation and preventing the development of autoimmune diseases [3].

During T-cell differentiation and activation, substrate requirement and energy metabolism change dramatically. Naïve T cells have low metabolic activity with minimal nutrient uptake, whereas during differentiation nutrient uptake is stimulated, the size of the mitochondrial pool is expanded and the energy requirement is mainly met by oxidative phosphorylation. By contrast, activated T cells primarily use aerobic glycolysis for ATP synthesis, a metabolic switch reminiscent of the so called “Warburg effect” [4]. The “Warburg effect”, named after German Nobel Prize laureate Otto Warburg, indicates the co-existence of vigorous glycolysis followed by a large lactic acid production in the presence of physiological O_2_ ambient levels (aerobic glycolysis). T-cell fate is also determined by changes in additional factors derived from the mitochondria or affecting their function, such as mitochondrial reactive oxygen species (mROS), mitochondrial matrix calcium, ATP, β-oxidation or other products, thus hinting to the key roles of mitochondria and their metabolic programmes in T-cell proliferation, differentiation and effector functions [5,6]. Intriguingly, we now know that the main product of mitochondrial energy metabolism, that is, ATP, is also an extracellular messenger released by virtually all cells, T cells included. Thus, the mitochondria participate in immunomodulation not only as sources of energy but also by producing a paracrine immunoregulatory factor, i.e., extracellular ATP, eATP. Thus, eATP can be considered for all intents and purposes a mitochondrially derived immunomodulatory messenger that links metabolism to immunity. Of course, eATP is not of exclusive mitochondrial origin as other messenger molecules, such as ROS, can be produced by additional pathways (glycolysis also generates ATP via substrate level phosphorylation in the cytoplasm), but without doubt the mitochondria are the main source.

## 2. Role of Energy Metabolism in T-Cell Activation and Proliferation

T cells enter the circulation from the thymus as naïve T cells, which are maintained in a quiescent G0 state by IL-17 stimulation [7]. Naïve T cells have low metabolic activity mainly centred on oxidative phosphorylation (OXPHOS) to support glucose-, glutamine- and fatty-acid-supported ATP generation [8]. When naïve T cells encounter their cognate antigen on an antigen-presenting cell (APC), the high-affinity T-cell receptor–major histocompatibility (TCR-MHC) peptide complex promotes the formation of an immune synapse (IS) and drives a change in metabolic programmes [9]. The mitochondria are relocated at the IS providing local ATP to maintain IS architecture and to sustain the intracellular calcium signalling cascade [10]. The local Ca^2+^ influx triggered by TCR stimulation increases the Ca^2+^ concentration in the cytosol and in the mitochondrial matrix with an overall critical effect on T-cell homeostasis, activation, differentiation and apoptosis. Moreover, extracellular Ca^2+^ influx and mitochondrial Ca^2+^ uptake are required for ROS production [11], as mROS generated by complexes I, II and III of the mitochondrial electron transport chain (ETC) fulfil a key second messenger role in the modulation of ROS-dependent transcription factors, such as nuclear factor of activated T cells (NFAT), nuclear factor-kB (NF-kB) and activator protein-1 (AP-1) [11]. T cells with a deficient complex III fail to activate NFAT transcription and the associated IL-2 production, but treatment with exogenous H_2_O_2_ is sufficient to recover IL-2 production [11]. The downregulation of complex I also impairs TCR-dependent ROS generation and the secretion of IL-2 and IL-4 [12]. In a mouse model of infection, antioxidant treatment reduced T-cell proliferation, further supporting the need for mROS in T-cell activation [13]. Of note, while low ROS concentrations support cell activation and proliferation, high levels are injurious and detrimental for cell viability. T cells producing increased levels of mROS due to their inability to prevent the formation of the mitochondrial permeability transition pore (mPTP) are highly susceptible to cell death [14]. The key role of the intracellular redox state in T-cell activation is also supported by the finding that the inability to keep a sufficient level of antioxidant glutathione (GSH) to buffer ROS prevents T-cell activation [15]. These observations suggest that titrating ROS levels is crucial for proper metabolic programming [16].

During T-cell exit from quiescence, TCR stimulation turns on energy metabolism to satisfy the increased bioenergetic and biosynthetic needs associated with proliferation, shifting from a quiescent state to a highly active anabolic state [17]. Mammalian target of rapamycin complex 1 (mTORC1) is the main regulator of quiescence exit through transcriptional, translational and post-translational mechanisms, but an important role is also played by mTORC2, which facilitates quiescence exit by activating AKT (protein kinase B) and up-regulating glucose transporter 1 (GLUT1) to enhance glucose uptake and glycolytic flux [18]. Upon antigen recognition, activated T cells increase aerobic glycolysis, the pentose phosphate pathway and glutaminolysis to support proliferation and maintain effector functions [19]. Metabolic reprogramming towards aerobic glycolysis and lactate production, the Warburg effect, is constantly associated with proliferating T cells, yet TCR stimulation dramatically increases mitochondrial mass and mitochondrial DNA levels, suggesting that the mitochondria may also play crucial roles in supporting T-cell activation and proliferation [20]. Mitochondrial respiration provides not only ATP but also building blocks for the anabolic processes supporting cell proliferation and growth. For instance, the NAD^+^-dependent conversion of malate into oxolacetate catalysed by mitochondrial malate dehydrogenase-2 (MDH2) is necessary for the synthesis of aspartate (a precursor of purine and pyrimidines) from oxalacetate. When the NAD^+^ pool is decreased due to compromised mitochondrial respiration, the de novo synthesis of aspartate is impaired, causing cell-cycle arrest [21]. Accordingly, MDH2 inhibition impairs T-cell proliferation [22]. Another mitochondrial anabolic pathway up-regulated during proliferation is glutaminolysis, in which glutamine is converted into α-ketoglutarate, which is then fed into the citric acid cycle (TCA). The production of α-ketoglutarate is thought to be a critical step, given that glutaminolysis inhibition impairs T-cell proliferation and cytokine production [23]. Overall, these observations support the importance of mitochondrial metabolism alongside glycolysis in T-cell proliferation.

## 3. Metabolic Reprogramming during T-Cell Differentiation

Activated T cells differentiate into different subsets depending on the given specific stimulated signalling pathway and the expression of downstream transcription factors. Each lineage displays a unique metabolic phenotype; however, the general assumption is that an increase in glycolysis boosts the differentiation of pro-inflammatory effector T cells (Teffs), whereas stimuli that increase mitochondrial metabolism boost the differentiation of regulatory T cells (Tregs) and CD8^+^ memory T cells [24] (Figure 1).

### 3.1. CD4^+^ T Cells

CD4^+^ T-cell subsets comprise pro-inflammatory Teff cells, among which there are Th1, Th2 and Th17, and suppressive Treg cells. Glycolysis is the main metabolic pathway for pro-inflammatory differentiation, and accordingly, Teff cells display elevated levels of glycolytic enzymes and lactate production [25]. GLUT1 down-modulation impairs whereas its overexpression increases glycolysis and Th1-, Th2- and Th17-cell proliferation [26]. Accordingly, the pharmacological inhibition of glycolysis by 2-deoxyglucose (2-DG) decreases Th17-cell differentiation and enhances Treg generation [27,28]. These data are further supported by the observation that mice deficient in hypoxia inducible factor-1α (HIF-1α), a key regulator of glycolysis in Teff cells, relied on OXPHOS and showed increased Treg-cell differentiation [27]. A recent study demonstrated that the activation of pyruvate dehydrogenase (PDH) through PDH kinase inhibition promoted Treg differentiation, whereas PDH inhibition (which causes increased glycolytic flux) boosted Th17-cell differentiation [25]. The administration of cell-permeant α ketoglutarate to CD4^+^ lymphocytes reprogrammed towards a mitochondrial metabolism, which skewed Treg cells towards a more pro-inflammatory phenotype [29]. Moreover, the administration of pan-antioxidant N-acetylcysteine (NAC) prevented Treg differentiation, suggesting that increased levels of mROS, on one hand, are necessary to drive Treg formation, and on the other, they may suppress Th17-cell differentiation [25]. It was suggested that mROS may indirectly modulate expression of forkhead box protein P3 (Foxp3), the transcriptional factor that promotes Treg differentiation by suppressing glycolysis-related genes and inducing the expressions of genes associated with lipid and oxidative metabolisms [30]. Cells expressing Foxp3 up-regulated all components of the ETC and enhanced fatty acid oxidation (FAO) [31], thus supporting the finding that Treg cells rely on FAO to fuel OXPHOS, which is required for maximal suppressive ability [32,33]. Accordingly, transforming growth factor-β (TGF-β) drove Treg differentiation [34] via AMP-activated protein kinase (AMPK)-dependent FAO, thus skewing inflammatory Th17 cells toward a Treg phenotype [24,35]. In addition, FAO inhibition using small-molecule drug etomoxir impaired Treg-cell differentiation, while FAO promotion by the pharmacological inhibition or genetic deletion of fatty acid synthase led to impaired Th17-cell differentiation and increased in vitro and in vivo Treg differentiation [24,36]. Moreover, the complex-I-mediated oxidation of NADH to NAD^+^ was essential for Treg suppressive function [33]. On the other hand, T cells with dysfunctional mitochondria become more glycolytic, differentiate into pro-inflammatory Th1 cells and secrete large amounts of IFN-γ [37]. These observations suggest that the balance between OXPHOS and aerobic glycolysis is crucial to ensure Treg lineage stability and support the hypothesis that mitochondrial metabolism is differentially required during T-cell subset differentiation.

Over the years, different hypotheses were put forward to explain why glycolytic T cells are more inflammatory than respiratory T cells. It was suggested that Th1-cell differentiation and IFN-γ production depend on an epigenetic mechanism that promotes an increased level of intracellular acetyl-CoA, which in turn leads to histone acetylation at loci that are crucial for Th1 differentiation and function, such as the locus encoding IFN-γ [38]. TCR stimulation was associated with the induction of the A chain of lactate dehydrogenase (LDHA), the enzyme that converts pyruvate to lactate. Lactate accumulation then indirectly promotes an increase in the acetyl-CoA cytoplasmic concentration. A recent study showed that the loss of the T-cell gene encoding the A chain inhibited Th1 cytokine production, owing to the lower cytosolic levels of acetyl-CoA [38]. Moreover, the glycolytic enzyme GAPDH post-transcriptionally down-modulated T-lymphocyte IFN-γ production by binding to cytokine mRNA in the AU-rich 3′ UTR region [39,40]. Recent evidence suggested that glutaminolysis besides glycolysis may also contribute to the inflammatory properties of Teff cells. Glutamine depletion or inhibition of glutaminolysis convert CD4^+^ T cells into Treg cells, even under Th1 polarizing conditions [41]. In addition, Th17 differentiation was associated with increased glutamine utilization, supporting the role of glutamine metabolism in the modulation of T-cell effector functions [42]. Finally, glycolytic inflammatory T cells are characterized by phosphoenolpyruvate accumulation, a metabolite that increases cytosolic Ca^2+^ levels promoting NFAT signalling and T-cell effector functions [43].

The metabolic reprogramming of CD4^+^ T cells is critically determined by mTOR signalling as mTOR activation promotes Th1- and Th17-cell differentiation, while mTOR inhibition induces Treg differentiation [44]. mTOR kinase exists in two different complexes: mTOR complex 1 (mTORC1) and mTOR complex 2 (mTORC2). mTORC1 is indispensable for Th1-cell differentiation, as T cells lacking the regulator of mTORC1 signalling (the GTP-binding protein RAS homolog enriched in the brain, Rheb) fail to differentiate into Th1 cells, even under Th1 polarizing conditions [45]. mTORC1 modulates cellular metabolism by regulating the expressions of the transcriptional regulators MYC and HIF-1α [46]. Transcription factor MYC regulates the expressions of several enzymes involved in glycolysis and glutaminolysis [47], while HIF-1α promotes Th17-cell differentiation by inducing master transcription factor RAR-related orphan receptor (ROR) γt through the aryl hydrocarbon receptor (AhR) pathway, but only in the presence of pro-inflammatory cytokines such as TGF-β, IL-6, IL-23 and IL-1β [48,49]. On the other hand, mTORC2 is essential for Th2-cell generation [45] by regulating the small GTPase RhoA. A recent study showed that RohA-deficient cells exhibited impaired Th2 differentiation and mTORC2 activity, while mTORC1 activity and Th1-cell polarization were preserved [50]. Accordingly, mTORC2 deficiency impaired Th2-, but not Th1- and Th17-cell differentiation both in vitro and in vivo [51]. Contrary to previous belief that CD4^+^-T-lymphocytes differentiation into various effector subsets is an irreversible event, Teff cells retain functional plasticity, and these cells can change their effector phenotype depending on the environmental conditions, for example, during infections, autoimmune diseases or allergic reactions [52,53].

### 3.2. CD8^+^ T Cells

CD8^+^ T cells are usually sub-classified into Teff cells, T memory cells or T effector memory cells, depending on their expression of specific genes and on the overall metabolic phenotype. Similarly to CD4^+^ T cells, an increase in the glycolytic flux and the up-regulation of glucose transporter GLUT1 are critical steps in CD8^+^-T-cell activation and differentiation and thus for proliferation and effector functions [54]. The correlation between glycolytic metabolism and CD8^+^-Teff-cell function is further supported by the observation that glycolysis is required for the rapid recall response of CD8^+^ memory T cells as they differentiate into secondary Teff cells [55,56]. An immediate-early glycolytic switch is required for rapid IFN-ɣ production, linked to mTORC1 activation and epigenetic modifications [57,58]. Costimulatory and coinhibitory pathways are also critical regulators of T-cell activation and metabolism. Activation through the TCR alone fails to induce metabolic reprogramming, while activation together with costimulatory molecule CD28 greatly increases glycolysis and mitochondrial metabolism through the phosphatidylinositol 3 kinase (PI3K)/Akt/mTORC1 pathway, enhancing T-cell function and proliferation [59]. This is also true of tumour-infiltrating CD8^+^ lymphocytes (TILs), where CD28 stimulation restores metabolism and functions by increasing glycolysis and mitochondrial activity [60]. Early metabolic studies showed that while glycolysis is a signal for effector functions, CD8^+^ memory T cells have a large capacity for OXPHOS and tend to be more quiescent [61,62,63]. CD8^+^ T cells treated with glycolysis inhibitor 2-DG show increased differentiation towards memory T cells, whereas the up-regulation of glycolytic enzymes decreases memory-T-cell differentiation [62]. Moreover, CD8^+^ memory T cells increase mitochondrial mass and have high spare respiratory capacity, largely owing to their increased engagement of fatty acid synthesis and FAO pathways [63].

During the life of T cells, mitochondrial morphology changes dynamically, and each change is associated with the engagement of distinct metabolic pathway. Mitochondrial dynamics, epitomized by changes in mitochondrial size, shape and cellular localization, were recently implicated in the control of T-cell fate. Upon activation, CD8^+^ Teff cells display fragmented mitochondria with loose cristae, a process known as fission, regulated by GTPase dynamin-related protein 1 (Drp1), which is responsible for the recruitment to the mitochondria of a protein complex inducing fragmentation [64]. In contrast to effector cells, CD8^+^ memory T cells have elongated mitochondria with tightly associated cristae as a result of the activity of protein Optic atrophy 1 (Opa1) that supports the fusion of the inner mitochondrial membrane and the proper maintenance of mitochondrial cristae, facilitating the assembly of the ETC complex and boosting OXPHOS activity [64,65] (Figure 1). The loss of Opa1 leads to defective CD8^+^-memory-T-cell generation, whereas CD8^+^ effector T cells treated with drugs that inhibit fission or promote fusion adopt a T-memory-like phenotype, reducing glycolysis and thereby activation [64]. All these findings suggest that not only mitochondrial metabolism but also mitochondrial dynamics control CD8^+^-T-cell differentiation and effector functions. A short summary of key metabolic pathways involved in CD4^+^ and CD8^+^ T-lymphocyte activation is reported in Table 1.

## 4. P2 Receptors in T-Cell Signalling and Metabolic Regulation

The expression of purinergic receptors by T cells was firstly suggested in 1978, when Gregory and Kern showed that eATP was mitogenic for mouse thymocytes [66]. Several years later, we confirmed these early findings by showing that eATP induced plasma membrane depolarization and permeability increase to low MW dyes and promoted the growth of human T lymphocytes [67,68]. About the same time, Sitkovsky and co-workers showed that TCR cross-linking caused ATP release, which in turn promoted T-cell effector functions via purinergic receptors [69]. Three families of purinergic receptors, P1, P2X and P2Y, are known. P1 receptors recognize adenosine and comprise four subtypes (A1, A2a, A2b and A3). P2X receptors (P2XRs) are ATP-gated ionotropic receptors permeable to small inorganic cations (Na^+^, K^+^ and Ca^2+^), consisting of seven members (P2X1-7) [70]. P2Y are metabotropic receptors (P2YRs) comprising eight members (P2Y_1_, P2Y_2_, P2Y_4_, P2Y_6_, P2Y_11_, P2Y_12_, P2Y_13_, P2Y_14_), gated by a wide range of nucleotide ligands [71]. Different combinations of these receptors are expressed by immune cells [72], but P2X1, P2X4, P2X7, P2Y_11_ and P2Y_12_ and emerged as key regulators of T-cell metabolism, migration and antigen recognition [73,74].

Over the years, it was conclusively demonstrated that autocrine/paracrine stimulation by released nucleotides (purinergic signalling) is an important mechanism of immune cell regulation [72,73,75,76,77]. Under basal conditions, T cells release a small portion of their intracellular ATP via vesicular exocytosis or ATP-permeable plasma membrane channels [78,79], but ATP release is quickly increased by the ligation of cell-surface receptors, such as the TCR. Regulated ATP release modulates several different cell responses; however, at sites of inflammation and tissue damage excessive amounts of ATP can accumulate with untoward effects on the local immune response [80,81]. Among the ATP-release pathways, pannexin-1 (panx-1) hemichannels, which are thought to be associated with the P2X7R and mediate TCR-dependent ATP release, are especially important in immune cells [82,83]. The inhibition of panx-1-mediated ATP release and the associated autocrine–paracrine P2X7R stimulation leads to decreased IL-2 secretion and T-cell proliferation, confirming that panx-1-mediated ATP release is a locally acting costimulatory signal [83]. Converging experimental evidence suggests that P2XRs are main targets for panx1-dependent autocrine-paracrine stimulation. In different non-immune cell types, the release of low amounts of ATP via panx-1 from resting cells overexpressing P2X receptors is sufficient to sustain the modest Ca^2+^ influx needed to maintain basal mitochondrial metabolism and ATP synthesis [84]. On this same line, P2X1Rs were shown to support energy metabolism in resting naïve CD4^+^ T cells by facilitating Ca^2+^ influx and sustaining the basal mitochondrial activity needed for T-cell vigilance [85]. Moreover, during T-cell activation, panx-1 hemichannels, mitochondria, P2X1R and P2X4R localize to the IS between the T cell and the APC (Figure 2). P2X1R and P2X4R trigger a localized Ca^2+^ influx that stimulates OXPHOS and propagates TCR-induced signalling, culminating in cytokine secretion and T-cell proliferation [86]. Junger and co-workers recently showed that mitochondrial trafficking to the immune synapse is directed by P2Y_11_Rs [87], suggesting that these receptors cooperate with P2X4Rs to recruit and stimulate mitochondria at the IS. P2X4Rs and P2Y_11_Rs fulfil similar roles in promoting T-cell migration. The autocrine stimulation of P2X4R induces cellular Ca2^+^ influx that supports the localized ATP synthesis required to promote the formation of those plasma membrane protrusions (pseudopods) needed to drive forward cell movement. The inhibition of P2X4R impairs T-cell polarization and migration in response to chemokine-receptor stimulation, pointing to the key role of these receptors in T-cell activation and migration [88]. While P2X4 receptors accumulate at the front of cells, P2Y_11_ receptors accumulate at the rear of polarized T cells, where they trigger cAMP/PKA signalling, which attenuates mitochondrial function at the back, preventing inappropriate pseudopod extension and promoting uropod retraction [89]. Altogether, these data suggest that P2Y_11_ and P2X4 receptors synergize to stabilize cell polarization and promote efficient T-cell migration. However, P2X4R/P2X7R ligation by eATP may also give a “stop” signal once immune cells have reached their destination, as, for example, lymphatic follicles [90].

Extracellular ATP is hydrolysed to ADP, AMP and adenosine by the ectonucleotidases expressed by different cell types, including T cells [91]. In T cells, ADP may activate specific P2YRs such as P2Y_1_Rs and P2Y_12_Rs [92]. In human CD4^+^ T cells, P2Y_1_Rs are upmodulated upon stimulation. Moreover, P2Y_1_R inhibition or silencing downregulates mitochondrial Ca^2+^ uptake, inhibits cytosolic Ca^2+^ signalling and, as a consequence, mitigates IL-2 production [93]. These findings demonstrate that P2Y_1_R activation participates in the autocrine purinergic signalling mechanism that boosts intracellular Ca^2+^ signalling and mitochondrial ATP synthesis in response to TCR stimulation. P2Y_1_Rs may also regulate T-cell migration as demonstrated by the finding that the inhibition of P2Y_1_R impairs Teff motility [93]. Although P2Y_12_R was thought to play a central role only in platelets, we now know that this receptor is also expressed by other cells of the immune system, such as monocytes and T cells, where it might promote proliferation, cytokine secretion, platelets–T-lymphocyte interaction and Treg differentiation and function in vitro, and in vivo immune response during sepsis [94,95,96]. These data support previous findings showing that P2Y_12_R antagonism or deficiency is linked to altered circulating cytokine levels in different animal models of inflammation [97,98,99].

P2Y_6_R signalling might also participate in the regulation of inflammation [100,101,102], although it is not yet clear whether its activation has a triggering or a protective function [103,104]. The lack of CD4^+^-T-cell P2Y_6_R increased cytokine generation and T-cell activation in an allergic lung model, suggesting that P2Y_6_R might have anti-inflammatory activity in the lung [101]. Accordingly, a recent study showed that intestinal inflammation in P2Y_6_-deficient mice was associated with an increased infiltration of macrophages, neutrophils, Th1 and Th17 cells and with the increased expressions of several inflammatory cytokines, such as IL-23, IL-6 and IL-1β [105].

### The P2X7 Receptor

In the P2XR family, the P2X7 subtype (P2X7R) plays an important role in the activation of T cells, mast cells, macrophages, dendritic cells and polymorphonuclear neutrophilic granulocytes [106,107,108]. The P2X7R is an ATP-gated ion channel associated with two different permeability states: a small-conductance cation-selective channel, and a large-conductance non-selective pore [109]. While following TCR stimulation other P2XRs such as P2X1R and P2X4R are rapidly translocated to the immune synapse, the P2X7R remains uniformly distributed on the plasma membrane, where it may function as a general sensor of eATP concentration, possibly to provide a trophic signal [86,110]. The role of P2X7R in T-cell activation was shown in human CD4^+^ T cells where P2X7R inhibition by different strategies, such as siRNA silencing or pharmacological blockade, decreased Ca^2+^ influx and inhibited T-cell activation [110], while genetic deletion in P2X7R-KO mice impaired leukocyte function and attenuated the inflammatory response [111]. Over the years, several reports highlighted the key role of the ATP/P2X7R axis in the modulation of Th17-cell polarization and differentiation [112,113,114]. P2X7R blockade inhibited Th17-cell differentiation and improved the outcome of heart and pancreatic islet experimental transplantation [113,115]. Positive effects were also observed in lupus-prone mice and in a study in myasthenia gravis patients, where the P2X7R blockade lowered the production of IL-17, IL-6 and IL-1β in isolated peripheral blood mononuclear cells [116]. P2X7R is differentially expressed by T-cell subsets; these, therefore, may be differentially susceptible to eATP-stimulated cell death [117,118,119]. In murine Treg cells, P2X7R could also be activated by ADP ribosylation catalysed by ADP-ribosyltransferase 2.2 (ART2.2) in the presence of NAD^+^ [120]. P2X7R activation by ART2.2 triggers calcium flux, phosphatidylserine exposure, L-selectin (CD62L) shedding, loss of the mitochondrial membrane potential and pore formation, resulting in cell death [62,120]. CD62L shedding, which is fundamental for T-cell migration, depends on the P2X7R-mediated activation of ADAM10 and ADAM17 metalloproteases, which also trigger CD27 and IL-6 receptor (IL-6R) shedding [121,122,123]. In human naïve T cells, CD26L shedding may depend on the eATP-mediated enhancement of mitochondrial superoxide generation [124]. Since CD62L promotes T-cell homing to secondary lymphoid organs, P2X7R may have a role in T-cell egress from lymph nodes, as suggested by findings showing that P2X4/P2X7 activation inhibited T-cell motility [90]. The P2X7R-mediated shedding of CD27, a member of the tumour necrosis factor family that supports antigen-specific expansion and T-cell memory generation [125,126], might be relevant for the regulation of adaptive immunity and/or immunopathology. On the other hand, the P2X7R-mediated shedding of IL-6R may skew T-cell polarization toward pro-inflammatory rather than immunosuppressive programmes.

The P2X7R also has a substantial role in the expansion, metabolic reprogramming and effector functions of CD8^+^ memory T cells [127]. P2X7R deficiency causes a defect in memory T cells ever since the initial precursor level differentiation stages, with a typical mitochondrial signature emphasized by lower mitochondrial mass and reduced spare respiratory capacity. Effects on CD8^+^-memory-T-cell differentiation due to the lack of the P2X7R may be mediated by AMPK/mTOR, since P2X7R-deficient T cells undergo an increase in intracellular ATP concentration, which inhibits AMPK activity. On the other hand, P2X7R activation triggers Ca^2+^ influx, which can activate AMPK. Finally, AMPK activity promotes memory differentiation by the inhibition of mTOR signalling [128]. P2X7R can also modulate ATP outflow through panx-1 channels, thus sustaining a memory-T-cell survival-promoting positive feedback loop between the P2X7R and panx-1channel fuelled by the constant efflux of intracellular ATP [129]. These data hint to the essential roles of energy metabolism, intracellular ATP and Ca^2+^ levels in promoting P2X7R-dependent T-cell differentiation.

Based on its immunomodulatory role, it is anticipated that an imbalance in P2X7R expression and activation may be implicated in the pathogenesis and progression of multiple diseases. The human *P2RX7* gene is highly polymorphic, with more than 150 non-synonymous single nucleotide polymorphisms (SNPs) (of which sixteen affecting receptor function were extensively characterized) being located in the extracellular loop or in the cytoplasmic tail [130]. Notably, some *P2RX7* SNPs are associated with various infectious, inflammatory and autoimmune diseases, such as tuberculosis [131,132], sepsis [133], multiple sclerosis (MS) [134] and rheumatoid arthritis (RA) [135]. The best characterized SNP is rs3751143, where adenosine (A) changes to cytosine (C) at position 1513 of the *P2RX7* gene. This modification causes a glutamic acid to alanine substitution at position 496 (E496A), generating a non-functional receptor [136]. ATP-induced IL-1β release in the whole blood from subjects harbouring the E496A SNP is lower than in subjects harbouring the wild type residue [137], suggesting that P2X7R loss-of-function SNPs may impair cytokine release by immune cells. The E496A polymorphism was associated with increased susceptibility to tuberculosis [138,139] and the less effective killing of other intracellular pathogens, such as *Toxoplasma gondii* [140]. In chronic Q fever, a persistent infection caused by the intracellular bacterium *Coxiella burnetii* [141], E496A may negatively affect the response to therapy [142]. On the other hand, the gain-of-function A348T (rs1718119) SNP, which is associated to increased IL-1β secretion from monocytes [143], is protective in several infectious diseases, such as toxoplasmosis [144], tuberculosis [145] and leprosy [146]. *P2R7X* SNPs may also substantially modify human T-cell, monocyte and NK-cell responses, as for example during graft-versus-host disease (GVHD) [147]. The expression of the loss-of-function E496A or I568N SNP in recipient subjects reduces post-HSCT (hematopoietic stem-cell transplantation) GVHD, highlighting the potential role of the *P2RX7* genotype in predicting GVHD onset [147,148]. *P2X7R* SNPs may also associate to the increased risk of autoimmune disorders characterized by a Th17/Treg disequilibrium [149,150], as P2X7R activation promotes Th1 and Th17 differentiation, decreasing Treg-cell proliferation.

Rheumatoid arthritis (RA) patients showed a significantly higher frequency of P2X7R-positive Th1 and Th17 cells and a decreased level of Treg cells compared to controls, suggesting that P2X7R expression is correlated with enhanced autoimmunity [151]. The A348T, Q460R and E496A SNPs were suggested to be susceptibility gene loci for RA [135,152], while the gain-of-function H155Y SNP is thought to contribute to RA pathogenesis due to increased P2X7R levels [153]. The gain-of-function H155Y and A348T SNPs were also implicated in the pathogenesis of systemic lupus erythematosus (SLE) [154,155], while P2X7R variants H155Y and A76V may have a role in the pathogenesis of multiple sclerosis (MS). A Spanish case-control study reported an increased frequency of P2X7R gain-of-function SNPs in MS, suggesting that an overall increase in P2X7R expression and functions may be a risk factor associated with its pathogenesis [134]. Interestingly, T cells isolated from the inflamed central nervous system in a model of MS had a distinct phenotype, suggesting that the priming site is a fundamental determinant of T-cell commitment to a defined T-helper-cell lineage with specific effector functions [156]. Notably, T helper cells derived from gut-draining mesenteric lymph nodes, which are primarily recruited to the white matter, expressed higher P2X7R levels than those derived from inguinal lymph nodes. These observations suggest that P2X7R may be a functional marker of effector T cells, and the characterization of the disease-associated SNPs in the *P2RX7* gene could help gain a better understanding of disease mechanisms. Moreover, since it is also released into the blood and its levels are significantly higher in diseased subjects [157], the P2X7R could be used as a disease biomarker.

An association of other P2R SNPs with selected diseases was found, for example, the *P2RY12* rs680969 SNP, which was proposed to predict resistance in child Kawasaki disease [158]. Other SNPs in *P2RY12* were suggested to be predictive of cancer pain severity [159] or to contribute to interindividual variability in the response to the widely used antithrombotic agent cangrelor [160]. Finally, the rs2305795 SNP in the 3′ untranslated region of the *P2RY11* gene was associated with narcolepsy [161]. An appraisal of the main disease-associated *P2X7R* SNPs is shown in Table 2.

## 5. Targeting T-Cell Metabolism and P2 Receptors: Potential Clinical Application

Inappropriate T-cell responses are the main causes of disease; therefore, dissecting the multiple pathways connecting T-cell metabolism, P2R signalling and immune function can very likely be helpful to devise new and more effective therapeutic strategies [162,163]. Autoimmune disorders share common features such as the generation of pathogenic effector CD4^+^ and CD8^+^ T cells. A direct link between autoimmunity and dysregulated T-cell energy metabolism was suggested by a large body of evidence. Naïve CD4^+^ T cells from lupus-prone mice exhibited higher glycolytic activity than those from controls [164], and the increased glucose uptake in mice overexpressing the GLUT1 transporter was by itself sufficient to drive IFN-ɣ and IL-2 production [165]. Deletion of GLUT1 reduced CD4^+^-T-cell glycolytic activity, and at the same time, it impaired effector CD4^+^-T-cell generation and mitigated the severity of colitis [26]. In addition, blocking glycolysis using 2-DG improved the clinical outcome in mice with experimental autoimmune encephalomyelitis (EAE) [27]. The antidiabetic hypoglycaemic drug metformin displayed anti-inflammatory properties. Treatment of a mouse model of SLE with a combination of metformin and 2-DG reversed metabolic changes and alleviated the disease [164]. Furthermore, statins (inhibitors of β-hydroxy β-methylglutaryl-CoA, HMG-CoA, reductase involved in cholesterol synthesis) seem to have a moderate immunosuppressive effect as they inhibited Th17-cell differentiation, ameliorating EAE and alleviating RA [166]. This is in line with the observations that fatty acid and cholesterol synthesis are required for T-cell proliferation. Amino acid restrictions can also affect T-cell activation and proliferation via the mTORC1 pathway [167]. Hence, limited nutrient availability can severely inhibit effector T-cell response and promote a tolerogenic environment [17]. While inhibiting T-cells response might be useful for the treatment of inflammatory and autoimmune diseases, the opposite approach should be used, in principle, for cancer treatment. The tumour microenvironment (TME) is characterized by low pH, low oxygen levels, low nutrient and high eATP concentrations, and other features that enforce dysfunctional T-cell metabolism and promote tumour immune tolerance and tumour progression [168]. Hypoxia and nutrient deficiency affect CD8^+^-T-cell responses by impairing fatty acids metabolism [169] and up-regulating various inhibitory receptors, such as programmed cell death-1 (PD-1) [170]. PD-1 stimulation severely affects cell metabolism by lowering the expression of the regulator of mitochondrial biogenesis peroxisome proliferator-activated receptor-gamma coactivator (PGC) 1-α, resulting in the reduction in glycolysis, the dysregulation of mitochondrial energetics and the increase in ROS production [170]. Promoting the expression of PGC1α or targeting the PD-1/PD-Ligand (PD-1L) 1 checkpoint with inhibitory antibodies restores mitochondrial function and increases anti-tumour responses [171,172]. PD-L1 is normally expressed by self-tissues to modulate peripheral tolerance but is also overexpressed by cancer cells, causing an impairment of T-cell function [173]. Recently, antibodies blocking cytotoxic lymphocyte antigen 4 (CTLA4) were also used in cancer therapy. CTLA4 is expressed on the surface of activated T cells, where its activation inhibits the PI3K/Akt/mTOR signalling pathway, impairing T-cell metabolism and proliferation [174,175,176] (Figure 3).

In addition to metabolites and metabolic enzymes [for example, phosphoenolpyruvate carboxikinase-1, PIM kinase (Proviral Integration site for Moloney murine leukemia virus, a highly conserved serine/threonine protein kinase) or acyl-CoA acyltransferase] [177], P2XRs are, in principle, useful targets for enhancing T-lymphocyte cytotoxicity in the TME [178], although due to their widespread expression also by tumour cells, the outcome of pharmacological P2XR stimulation or inhibition is difficult to anticipate. Biological molecules or chemical compounds acting as positive allosteric modulators of the P2X7R (the bactericidal peptide LL-37, the antibiotic polymixn B, the H1 histamine antagonist clemastine or the anti-inflammatory compound Tenidap), or of the P2X4R (the anti-helminthic drug Ivermectin) could be used to potentiate eATP-mediated T-lymphocyte activation, thus promoting a more vigorous anti-tumour response [179,180,181,182,183]. These compounds may also synergize with the elevated eATP levels typical of the TME to trigger cancer cell death and to promote an anti-tumour protective response. On the other hand, P2X7R-expressing immune cells in the TME may also be harmed by these positive allosteric drugs, thus impeding immune-cell responses. Therefore, the net effect of P2X7R stimulation/blockade on cancer progression is not as yet clear and very likely depends on given specific TME and the specific T-lymphocyte subset. Grassi and co-workers recently reported that the lack of P2X7R inhibited TIL senescence and enhanced cytotoxic-T-lymphocyte responses in mouse solid tumours [184]. In a different tumour model, Adinolfi and co-workers showed that the P2X7R blockade increased CD4^+^-T-cell infiltration, without affecting the CD8^+^ and Treg populations. Moreover, in this model, P2X7R-inhibited CD4^+^ T cells down-modulated CD39 and CD73 ectonucleotidases, suggesting that P2X7R blockade might promote the generation of a less immunosuppressive TME [185]. While no clinical trials in cancer have been ran so far, the P2X7R blockade was explored in a few chronic inflammatory diseases in over 20 Phase I/II trials [186,187]. These trials showed an excellent tolerability of the P2X7R blockade, but efficacy was not sufficient to warrant further clinical development. An interesting therapeutic application of P2X7R-targeted reagents might be to deplete P2X7R-overexpressing Treg cells, thus potentiating the anti-tumour immune response. Koch-Nolte and co-workers showed that the in vivo administration of NAD^+^ caused a dramatic Treg depletion and tumour regression in EL-4 or EG7 lymphomas, MCA fibrosarcoma and B16 melanoma mouse models [188]. More recently, specific anti-P2X7R nanobodies (small antibodies derived from those found in camelids) were generated with either P2X7R-blocking or -enhancing activity, potentially useful in the deletion of P2X7R-expressing cells or in the tuning of the P2X7R-dependent responses in cancer as well as inflammation [189].

Few clinical applications are reported for compounds active at other P2 receptors, besides the long-known and highly successful P2Y_12_R inhibitors (Prasugrel, Clopidogrel, Cangrelor, Ticagrelor), widely used for the treatment of thrombotic diseases [190,191]. Two P2Y_2_R agonists (diquafasol and denufosol) for the treatment of dry eye disease, a P2X3R antagonist (gefapixant) and a P2X4R antagonist (NC-2600) for the treatment of chronic, non-productive, cough were licensed (see [192] for review). Recently, the P2XR agonist αβ-methylene ATP (αβ-ATP) was demonstrated to act as a mucosal cancer vaccine adjuvant, leading to an enhanced antigen-specific cytotoxic-T-lymphocyte (CTL) response [193], lithocholic acid was shown to potentiate the P2X4R [194], and aurintricarboxylic acid was identified as a potent allosteric inhibitor of P2X1R [195], but no additional information is available.

## 6. Conclusions

Thus, in conclusion, P2 receptors are emerging as potent modulators of T-lymphocyte metabolism, differentiation and effector functions. Their expression is associated with several widely diffused human pathologies, and a number of SNPs affecting receptor functions have been identified, thus underscoring their potential as novel markers of disease.

## Figures and Tables

**Figure 1 biomolecules-12-00983-f001:**
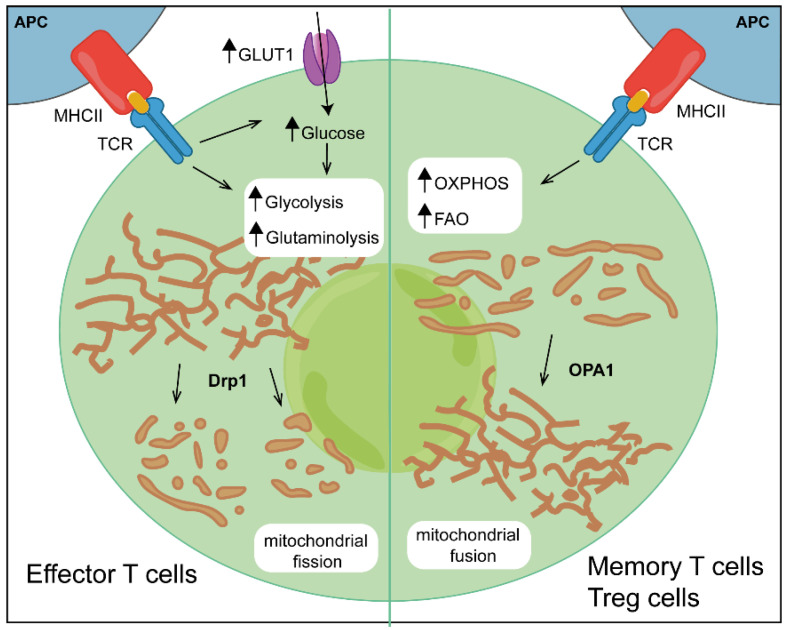
Mitochondrial metabolism and dynamics determine T-cell differentiation. Increasing aerobic glycolysis and glutaminolysis leads to a pro-inflammatory effector phenotype, whereas stimuli that promote OXPHOS and mitochondrial respiration boost the differentiation of regulatory and memory T cells. Morphological changes in the mitochondria also control metabolic reprogramming: the mitochondria of effector T cells undergo fission regulated by the GTPase dynamin-related protein 1 (Drp1), while regulatory and memory T cells maintain a fused mitochondrial network supported by protein Optic atrophy 1 (Opa1). FAO, fatty acid oxidation.

**Figure 2 biomolecules-12-00983-f002:**
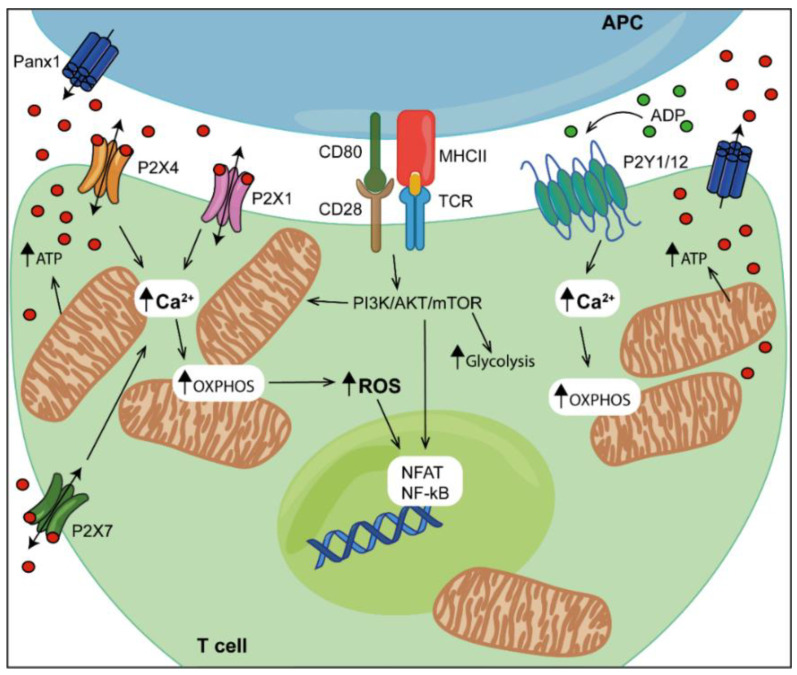
P2 receptors and mitochondria concur to support T-cell activation. Ligation of both T-cell receptor (TCR) and the costimulatory molecule CD28 increases glycolysis and mitochondrial metabolism through the phosphatidylinositol 3 kinase/Akt/mammalian target of rapamycin complex 1 (PI3K/AKT/mTOR) pathway and the downstream activation of transcription factors such as nuclear factor of activated T cells (NFAT) and nuclear kappa-light-chain-enhancer of activated B cells (NFkB). Upon antigen recognition, mitochondria are relocated to the immune synapse to sustain calcium signalling and supply local ATP, which is also partially released via pannexin 1 (panx-1) hemichannels. Extracellular ATP activates P2X1R, P2X4R and P2X7R, triggering a localized calcium influx that stimulates oxidative phosphorylation (OXPHOS) and mitochondrial respiration. Extracellular ATP is hydrolysed to ADP, which activates specific P2Y receptors such as P2Y_1_R and P2Y_12_R, boosting intracellular calcium signalling and mitochondrial ATP synthesis. Moreover, extracellular Ca^2+^ influx and mitochondrial Ca^2+^ uptake are required for reactive oxygen species (ROS) production and the modulation of ROS-dependent transcription factors NFAT and NF-kB. APC, antigen-presenting cell; panx-1, pannexin-1.

**Figure 3 biomolecules-12-00983-f003:**
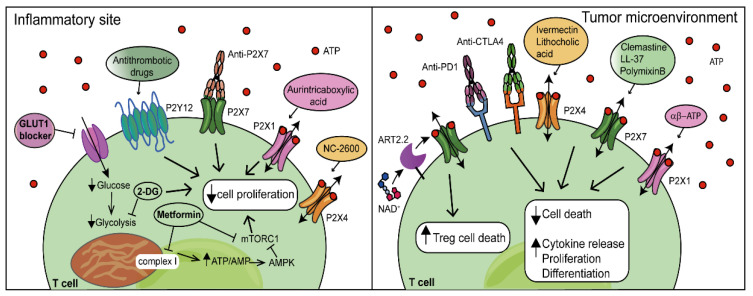
Targeting cell metabolism and P2 receptors to modulate T-cell activation and differentiation. Metabolic inhibitors such as 2-deoxyglucose (2DG), metformin or glucose transporter (GLUT) 1 blockers inhibit T-cell responses and reduce proliferation, providing potentially useful targets for the treatment of inflammatory and autoimmune diseases. On the contrary, improving T-cell metabolism and activation by inhibiting programmed cell death (PD)-1 or cytotoxic lymphocyte antigen (CTLA) 4 receptors is a current cancer treatment option. In addition, P2 receptors are emerging as useful targets to inhibit or enhance T-cell metabolism and effector functions at inflammatory sites or in the tumour microenvironment, respectively. 2-DG, 2-deoxyglucose; mTORC1, mammalian target of rapamycin complex 1; AMPK, AMP-activated protein kinase; ART2.2, ADP-ribosyltransferase 2.2.

**Table 1 biomolecules-12-00983-t001:** Metabolic regulators of T-cell differentiation.

	CD4^+^ (Th1, Th2, Th17) and CD8^+^ Effector T Cells	Treg Cells Memory CD8^+^ T Cells
**Primary source of energy**	GlycolysisGlutaminolysis	Fatty acid oxidation (FAO)OXPHOS
**Transporter**	Amino acid transporterGlucose transporter (GLUT1)	Low nutrient uptake
**Kinases**	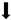 AMPK 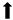 mTORC1/2	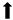 AMPK 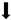 mTORC1/2
**Transcription factors**	MycHIF-1α	HIF-1αFoxp3

**Table 2 biomolecules-12-00983-t002:** *P2RX7* SNPs associated with inflammatory and autoimmune diseases.

dbSNP ID	Nucleotide Change	Amino Acid Change	Effect on Functions	Implicated Conditions
rs3752243	1513 A > C	E496A	Loss	Tuberculosis [137,138] *Toxoplasma gondii* infection [139] Chronic Q fever [140,141] Graft-versus-host disease (GVHD) [146] Rheumatoid arthritis [151]
rs1718119	1068 G > A	A348T	Gain	Toxoplasmosis [142] Tuberculosis [143] Leprosy [144] Rheumatoid arthritis [134] Systemic lupus erythematosus [153]
rs1653624	1729 T > A	I568N	Loss	Graft-versus-host disease (GVHD) [146]
rs2230912	1405 A > G	Q460R	Partial loss	Rheumatoid arthritis [151]
rs208294	489 C > T	H155Y	Gain	Rheumatoid arthritis [152] Systemic lupus erythematosus [154] Multiple sclerosis [133]
rs17525809	370 T > V	A76V	Gain	Multiple sclerosis [133]

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
