# Peer review of "P2 Receptors: Novel Disease Markers and Metabolic Checkpoints in Immune Cells"

_biomolecules, 2022, doi:10.3390/biom12070983_

Round 1

Reviewer 1 Report

In this article, Vultaggio-Poma and Di Virgilio provide an extensive overview on the metabolic changes occurring during T cell activation and differentiation, and the role of P2 receptors in this context. The authors have undoubtedly done a tremendous effort in reviewing the literature on the topic, which is highly appreciated. However, the resulting manuscript is difficult to follow, reiterative at times, and in occasions takes for granted some concepts that might be not so obvious for readers that are not specialized in this specific research field. Thus, the authors are strongly encouraged to make a thorough review of the manuscript and make an effort to present all this information in a more reader-friendly way. Here there are some suggestions that might help towards this aim:

1. Review all the acronyms that are used along the manuscript and make sure that they are properly explained the first time they are mentioned. For example, in line 44 oxidative phosphorylation is mentioned for the first time, but its acronym (OXPHOS) is not included. Then, in line 63, we find the acronym OXPHOS without any further explanation. Similarly, the acronym FAO appears the first time in Figure 1, line 121. However, we do not know what FAO means until line 151. There are several examples like these ones, so please, make sure that they are properly corrected.

2. Some sections are extremely dense and feel like a list of unconnected facts, which sometimes results in reiterative statements. For example, in section 2, line 66, it is mentioned for the first time that TCR stimulation drives metabolic changes. And only 22 lines later (lines 88 through 91), this very same fact is stated again. Please, make an effort to structure the information in a more organized and digestible way. It might help to further subdivide sections in several subsections and support them with more illustrative figures. For example, and continuing with case of section 2, maybe this section could be subdivided in two subsections, namely the role of ROS and the metabolic switch. Also, some of the information could be presented in the form of tables instead of text, which would significantly lighten up manuscript. For example, the part regarding the P2X7R SNPs could definitely benefit from this kind of approach.

3. As mentioned before, there are concepts that are taken for granted by the authors, which might not be so obvious for certain readers and should be explained, such as “purinergic signaling”, “Warburg effect”, or “pseudopod/uropod”, among others.  Also, it is important to make clear in which cell compartments occur the different metabolic processes. It is never mentioned that aerobic glycolysis occurs in the cytoplasm. Moreover, the title of Figure 1 is “Mitochondrial metabolism and dynamics decide T cell differentiation” and the first metabolic pathway that is mentioned in that figure legend is aerobic glycolysis, which might mislead the reader towards thinking that aerobic glycolysis occurs in the mitochondria.

4. Also, in Figure 1, using those brown filaments to represent the mitochondria is not very illustrative. They look rather like the Golgi apparatus or the endoplasmic reticulum, which is very confusing.

5. Please, review also the text for typos and punctuation errors, such as missing commas (line 36 after “Th2 cells”, line 79 after “In a mouse model of infection”) or misspelled words (line 169 “aceTyl-CoA”).

6. Finally, the text is riddle with grammatical errors. Please, make sure that a native or proficient English speaker proofreads the manuscript before resubmission.

Reviewer 2 Report

The work is centered on an interesting topic and information rich. In general English usage is fine.

I noted that ATP is considered an autocrine signal by the authors and the majority of literature. However, evidences of paracrine functions exists (Cell Death Dis. 2014 Mar; 5(3): e1102; Shock. 2014 Aug; 42(2): 142–147) this may be considered especially in the context of active lymph nodes. Some articles suggest a different function of purinergic signaling in migration reduction once the lymphocyte arrives in lymph nodes “EMBO J. 2014 Jun 17;33(12):1354-64. Front Immunol. 2021 Dec 22;12:786595.” please discuss.

Minor points:

There are several fragments / confused sentences that call for revision:

Row 43 “whereas during nutrient uptake is stimulated,”

Row 63 “OXPHOS to generate glucose-, glutamine- and fatty acid-fuelled ATP generation”

Row 270 “Among ATP-release pathways, pannexin 1 (panx1) hemichannels are especially important in immune cells” tray to explain shortly about hemichannels and regulation of ATP release by TCR.

Suggestion to be meet at author discretion:

Chapters on CD4+ T cells and CD8+T cells are very dense with information a scheme/graph may help the reader and attract citations.

Reviewer 3 Report

Vultaggio-Poma and colleagues describe the role of eATP and purinergic receptors in T cells. The review is well structured and nice to read. I have few suggestions to improve the text.

At lines 131-165 authors describe how glycolysis and OXPHOS is regulated in CD4 T cell subsets. It could be nice to add to this discussion a comment about a recent work by Dardalhon and colleagues (10.1016/j.celrep.2021.109911) who demonstrate that increasing aKG levels impairs Treg differentiation by altering ETC-II activity and epigenetics and showing a more complex relationship between OXPHOS/glycolysis and th17/th1/Treg cells

The interpretation of reference [37] is incorrect. The authors said at line 171-173 “TCR stimulation is associated with the induction of the A chain of lactate dehydrogenase (LDHA), the enzyme that converts pyruvate to lactate. Lactate is then converted to acetyl-CoA.” Lactate is not converted into acetyl-CoA unless it goes through pyruvate. In the reference [37] author actually demonstrate that LDHA by regenerating NAD+ allows glycolysis to generate ATP at higher rate. This in turn allows mitochondria to reduce ATP production (since glycolysis can now compensate for it) and reduce the TCA cycle rate. Citrate can then be exported into cytosol where it is converted to acetyl-CoA by ACLY. Therefore, LDHA promotes ac-CoA accumulation but indirectly and not through a direct conversion of lactate into acetyl-CoA.

Round 2

Reviewer 1 Report

This reviewer appreciates the effort of Vultaggio-Poma and Di Virgilio in providing an improved and more reader-friendly version of their manuscript. The concerns regarding the first version have been mostly and satisfactorily addressed. The only point I must insist on is the fact that the representation of mitochondrias in figure 1 is still very misleading. The authors should consider using a representation like the one they use in figure 2, which is way more illustrative than the filaments/formless shapes used in figure 1. Other than that, the article is very educational and of general interest in the field.

Author Response

Thanks

Reviewer 2 Report

My criticism were addressed by authors.

Author Response

Thanks